# Non-Tectonic Geohazards of Guangdong Province, China, Monitored Using Sentinel-1A/B from 2015 to 2022

**DOI:** 10.3390/s24165449

**Published:** 2024-08-22

**Authors:** Jincang Liu, Zhenhua Fu, Lipeng Zhou, Guangcai Feng, Yilin Wang, Wulinhong Luo

**Affiliations:** 1Surveying and Mapping Institute Lands and Resource Department of Guangdong Province, Guangzhou 510630, China; liujincang@gdsgtzychy.wecom.work (J.L.); zhouli-peng@gdsgtzychy.wecom.work (L.Z.); 2The Key Laboratory of Natural Resources Monitoring in Tropical and Subtropical Area of South China, Ministry of Natural Resources, Guangzhou 510670, China; 3School of Geosciences and Info-Physics, Central South University, Changsha 410083, China; fredgps@csu.edu.cn (G.F.); 235011017@csu.edu.cn (Y.W.); 235011031@csu.edu.cn (W.L.)

**Keywords:** Guangdong Province, geological hazards, multi-temporal InSAR, surface subsidence, Sentinel-1

## Abstract

Guangdong Province, home to 21 cities and a permanent population of 127.06 million people, boasts the largest provincial economy in China, contributing 11.76% to the national GDP in 2023. However, it is prone to geological hazards due to its geological conditions, extreme weather, and extensive human activities. Geohazards not only endanger lives but also hinder regional economic development. Monitoring surface deformation regularly can promptly detect geological hazards and allow for effective mitigation strategies. Traditional ground subsidence monitoring methods are insufficient for comprehensive surveys and rapid monitoring of geological hazards in the whole province. Interferometric Synthetic Aperture Radar (InSAR) technology using satellite images can achieve wide-area geohazard monitoring. However, current geological hazard monitoring in Guangdong Province based on InSAR technology lacks regional analysis and statistics of surface deformation across the entire province. Furthermore, such monitoring fails to analyze the spatial–temporal characteristics of surface deformation and disaster evolution mechanisms by considering the local geological features. To address these issues, current work utilizes Sentinel-1A/B satellite data covering Guangdong Province from 2015 to 2022 to obtain the wide-area surface deformation in the whole province using the multi-temporal (MT) InSAR technology. Based on the deformation results, a wide-area deformation region automatic identification method is used to identify the surface deformation regions and count the deformation area in each city of Guangdong Province. By analyzing the results, we obtained the following findings: (1) Using the automatic identification algorithm we identified 2394 deformation regions. (2) Surface subsidence is concentrated in the delta regions and reclamation areas; over a 4 cm/year subsidence rate is observed in the hilly regions of northern Guangdong, particularly in mining areas. (3) Surface deformation is closely related to geological structures and human activities. (4) Sentinel-1 satellite C-band imagery is highly effective for wide-area geological hazard monitoring, but has limitations in monitoring small-area geological hazards. In the future, combining the high-spatial–temporal-resolution L-band imagery from the NISAR satellite with Sentinel-1 imagery will allow for comprehensive monitoring and early warning of geological hazards, achieving multiple geometric and platform perspectives for geological hazard monitoring and management in Guangdong Province. The findings of this study have significant reference value for the monitoring and management of geological disasters in Guangdong Province.

## 1. Introduction

Guangdong Province in China is highly populated and has several major cities like Guangzhou and Shenzhen. Any geological disaster in this region can have devastating consequences, affecting millions of people and causing severe damage to the economy. However, this province is prone to various geohazards, such as landslides, debris flows, floods, and subsidence, due to geological structures and extreme weather [1]. Until 2020, 4744 geohazards had been recorded, including 2646 collapses (56%), 1654 landslides (35%), 77 debris flows (2%), 94 cases of ground subsidence (1%), 24 cases of ground settlement (1%), 10 ground fissures, and 239 unstable slopes (5%) [2]. These hazards have caused economic loss as huge as RMB 8.18 billion and affected the life of 249,500 people [2]. Understanding the distribution of these hazards is crucial for developing effective and sustainable solutions to mitigate the risks posed by geohazards [3,4].

Traditional ground subsidence monitoring methods rely on point observations, such as leveling measurements, total station measurements, and GNSS measurements. While these methods offer high precision, they have limited coverage, poor timeliness, and high operational costs, making them inadequate for wide-area geological hazard surveys and rapid monitoring. In addition, the weather in Guangdong Province is often cloudy and rainy, which is unfavorable for traditional deformation monitoring methods. Interferometric Synthetic Aperture Radar (InSAR) technology enables all-weather, wide-area surface deformation monitoring with centimeter- to millimeter-level accuracy [5,6]. Therefore, InSAR observations may provide satisfactory geohazard monitoring in those regions [7,8].

With the development of InSAR satellite technology and the increasing number of images, the time-series InSAR techniques such as Persistent Scatterer InSAR (PS-InSAR), Quasi-Persistent Scatterer InSAR (QPS-InSAR), Small Baseline Subset InSAR (SBAS-InSAR), and Distributed Scatterer InSAR (DS-InSAR), have been widely applied in disaster monitoring and management in Guangdong Province [9,10,11]. Many geological hazard surveys and studies in Guangdong have been carried out on the central and southern regions, as well as the Pearl River Delta. These areas are economic centers, characterized by high urbanization, a dense population, and extensive underground infrastructure [12,13,14]. The geological hazards in these regions are ground subsidence and collapse. Some studies used InSAR technology to monitor the deformation of linear man-made features like highways, railways, urban subways, underground pipelines, bridges, and embankments, during construction and operation [15,16,17]. In addition to precise deformation monitoring of structures, InSAR has great advantages in extensive surface deformation monitoring. Some scholars have conducted wide-area subsidence monitoring in cities, coastal zones, and delta regions of Guangdong [18,19,20,21]. They considered local geological environmental characteristics in the analysis of the causes of ground subsidence, revealing mechanisms, and contributing to disaster management.

The research on geological hazard monitoring in Guangdong Province using InSAR technology has yielded some promising results [17,19,22], but there are several challenges: (1) Most studies focused on densely populated areas with noticeable subsidence and deformation, such as the Pearl River Delta and Hanjiang Delta. Few systematic studies analyzing hilly regions like northern Guangdong and western Guangdong, which are prone to geological hazards, have been reported. Furthermore, analysis of the surface deformation across the entire Guangdong Province is rather rare. (2) No analysis has been carried out between geological hazard mechanisms and the spatial–temporal characteristics of surface deformation on the temporal and spatial domains. These limitations hinder a deeper understanding of the drivers of geohazards in Guangdong.

In this study, we use Sentinel-1A/B data from 2015 to 2022 to conduct a comprehensive survey of the geological hazards throughout Guangdong Province and analyze their causes. The study intends to investigate the spatial–temporal characteristics of geological hazards, providing technical support for disaster management and early warning. Firstly, employing the time-series InSAR technology, we process all available Sentinel images during the study period, obtaining high-spatial-resolution (40 m) surface deformation data for the entire province. Based on the deformation information, we study the spatial distribution characteristics of geological hazards in Guangdong. Furthermore, we utilize an automatic identification method to identify the deformation regions in each city of the province and calculate the area of these regions. Finally, we discuss the correlation between disasters and human activities, along with the capabilities of C-band Sentinel images in monitoring disaster surveys in Guangdong Province.

## 2. Geological Structure and Geological Hazard Background

### 2.1. Geological Structure and Climatic Environment

Geological movements, such as the Yanshan and Xishan events, formed three major fault systems: the NE (north-east), NEE (north-east-east), and NW (north-west) faults, along the coast of Guangdong Province, forming the fundamental development of and trends in the coastal landforms. Geological hazards in Guangdong Province are primarily affected by fault systems, especially the NE fault. In the eastern Guangdong fold that has areas of low hills formed by erosion and denudation and the western Guangdong fault block that has areas of low hills formed by uplift and erosion, landslides and collapses are the main geological hazards. Regions with karst development in northern Guangdong are prone to ground subsidence. The fault depression accumulation plains of the Pearl River Delta and fault depression erosion accumulation tablelands of Leizhou Peninsula experience frequent ground subsidence. Earth fissures are primarily found in the fault depression erosion accumulation tableland area of Leizhou Peninsula.

Guangdong Province is located in the southern tip of mainland China, with a coastline length of 3368.1 km, and it has numerous islands and harbors. The topography is higher in the north and lower in the south. The northern region is predominated by mountainous and hilly terrain. The low mountains include the Lianhua Mountain and the Yunwu Mountain. The terrain gradually transitions to mountains and plains in the south. The largest plain in the south is the Pearl River Delta, a sediment plain located at the estuaries of the West River, North River, and East River, characterized by the widespread thick soft-soil layers. Guangdong Province falls within the East Asian monsoon region, featuring a warm and humid climate with ample sunshine. The annual average temperature is high, and rainfall is concentrated from April to October, aligning with the highest temperatures. During the rainy season, precipitation accounts for approximately 70% to 85% of the annual total. Rainfall distribution across the region is uneven, showing a multi-center pattern. The three main rainfall centers are Enping, Haifeng, and Longmen (as shown in Figure 1). These climatic conditions make Guangdong Province susceptible to heavy rainfall, flooding, typhoons, drought, and other meteorological disasters [23].

### 2.2. Geological Hazard Types and Distribution

The geographical conditions determine the main distribution characteristics of geological hazards, and human activities affect the distribution. The terrain in Guangdong Province includes mountains, hills, and plains. Generally, landslides and collapses frequently occur in hilly and plateau edge areas, due to steep terrain and poor slope stability. Yangjiang, Haifeng, and Leizhou are prone to landslides and collapses. Carbonate rocks are widely distributed in the mountainous regions of northern Guangdong, the Guanghua Basin, Yangchun, and Zhaoqing, where regional subsidence often occurs, due to karst development. In the central and southern coastal areas of the Pearl River Delta, thick soft-soil layers are widespread and shallowly buried, so ground collapse and subsidence are the main geohazards. Situated in the southern coastal monsoon region, Guangdong experiences abundant rainfall throughout the year and is influenced by ocean currents. Heavy rain coupled with floods often lead to mudslides and landslides. Since the 1980s, Guangdong Province has seen significant economic development, bringing increasingly intense human activities. In the southern region, significant disruptions to the environment have led to more frequent occurrences of landslides, collapses, ground subsidence, and other geological hazards. These hazards pose a severe threat to the safety of people’s lives and property, as well as the social and economic sustainable development of the region.

According to official records [2], the province has 18 high-risk geological hazard areas and 17 moderate-risk areas, covering an area of 104,600 km^2^, which accounts for 58.1% of the total land area of the province (Figure 1a). From 2016 to 2020, a total of 1155 geological hazards occurred, with 97.2% small-scale incidents. By the end of 2020, Guangdong Province had identified 4744 geological hazard risk points, including 48 extra-large, 426 large, 2078 medium, and 2192 small points, as shown in Figure 1b. As for hazard types, these disasters include 2646 landslides, 1654 rock-slides, 77 mudslides, 94 ground collapses, 24 ground subsidence areas, 10 ground fissures, and 239 unstable slopes (Figure 1c).

As Figure 1 shows, the geological hazards in Guangdong Province are characterized by landslides and rockslides, mainly occurring in the hilly areas in the northern part of the Pearl River Delta, as well as in the mountainous regions of western, northern, and northeastern Guangdong. In the hilly areas of the Pearl River Delta, high population density and economic development bring frequent human engineering activities, leading to ground subsidence. In the western, northern, and northeastern areas of Guangdong, where flatlands are limited, people cutting into slopes for construction is normal, which coupled with less economic development, leads to a lack of slope stabilization. Therefore, analyzing the causes and influencing factors of these disasters is of significant importance for future geological hazard prevention and control.

## 3. InSAR Data Processing and Deformation Analysis

### 3.1. InSAR Dataset Introduction

To obtain the wide-area surface deformation of Guangdong Province, we collected the SAR images covering the study area acquired by the C-band Sentinel-1 satellite from June 2015 to December 2022. Due to the lack of the descending orbit in Sentinel-1 data in Guangdong Province, we only acquired the ascending orbit data. The Sentinel-1 data were collected in the satellite’s default observation mode of Interferometric Wide Swath (IW). We obtained approximately 2000 images from 11 frames in 5 tracks. The swath width of the image is 250 km, with a ground resolution of 5 m × 20 m (range × azimuth). The image coverage is shown in Figure 2, and the detailed data parameters are listed in Table 1.

### 3.2. Multi-Temporal InSAR Technology

The data of study area are processed by GAMMA (v1.1 11-May-2023) software. We used a time-series SBAS InSAR technique based on multi-master images to obtain the surface deformation in Guangdong Province from June 2015 to December 2022. A total of approximately 2000 images from 5 tracks and 11 frames, comprising about 6000 differential interferograms, were processed [24]. The data processing procedure is as follows: (1) Data preprocessing. First, each frame’s images were registered with a registration accuracy better than 0.001 pixels. Due to the short spatial baselines between Sentinel-1 images, only the temporal baseline was considered for selecting interferometric pairs. The temporal baseline threshold was set to 60 days [25]. Interferometric pairs were formed based on the principle of connecting each image with the two subsequent images in time. Before phase unwrapping, the threshold for the coherence mask is set between 0.1 and 0.9. One of the interferogram network figures formed by one track is shown in Figure 3. (2) Differential interferometry. The selected interferometric pairs were processed with a differential interferometry ratio of range-to-azimuth of 10:2 (ground resolution of about 40 × 40 m) to reduce noise and data volume. The baseline data were used to remove the flat-earth phase, and then the 30 m resolution SRTM DEM data were used to simulate and remove the topographic phase from the interferograms, resulting in differential interferometric phases. The adaptive Goldstein filtering method was applied to the differential interferometric phase to reduce the phase noise, followed by phase unwrapping using the minimum cost flow method to obtain the unwrapped differential interferometric phase. The unwrapped phase contains deformation signals as well as errors such as orbital errors, atmospheric delay errors (mainly tropospheric delay), residual topographic errors, and noise. Polynomial surface fitting was used to remove orbital errors. Singular Value Decomposition (SVD) was used to calculate the initial deformation sequence and residual topography [26]. (3) Time-series deformation solution. The initial deformation sequence was subjected to linear regression analysis over time to obtain linear deformation. The linear deformation was then removed from the initial deformation sequence to obtain residual phases. High-pass filtering in the time domain is applied to remove the atmospheric phase and noise, isolating the nonlinear deformation. Finally, the linear and nonlinear deformations were summed to obtain the final time-series deformation results. (4) Deformation results in adjustment and splicing. To ensure the deformation reference baseline consistency and deformation results in continuity between different image frames, a least-squares-based splicing method proposed by Wang et al. [27] was employed to splice the results. The results of different frames were corrected by the global adjustment model, which uses the spatial consistency of homonymy points in overlapping regions [28]. Finally, the results were spliced to obtain a wide-area time-series deformation map covering the study area.

### 3.3. Surface Deformation Results and Field Verification

Using the method described in Section 3.2, we obtained the deformation velocity map in every path and frame. To ensure a consistent project system for surface deformation, we converted the LOS deformation into the vertical direction [29]. The deformation velocity map of Guangdong Province from 2015 to 2022 is shown in Figure 4. The surface subsidence in Guangdong Province exhibits the following features: surface subsidence concentrates in the delta fronts and coastal zones, with some significant deformation occurring in northern Guangdong, mainly in the Leizhou Peninsula, Pearl River Delta, and Hanjiang Delta. Subsidence velocities range between −60 and −10 mm/year, with regional subsidence rates exceeding −70 mm/year. Subsidence in coastal areas is sporadic and is mainly located in the reclaimed land areas and aquaculture zones. In some mining areas in northern Guangdong, subsidence occurs around the mining sites, with subsidence rates ranging from −30 to −10 mm/year, and a few regions exceeding −40 mm/year.

The uncertainty of the deformation velocity map of Guangdong Province is presented in Figure 5, which reveals the error distribution range of the calculated deformation results. It is notably observed that the uncertainty in most areas remains below 1 mm/year, indicating the high accuracy of the overall deformation results. However, it is noteworthy that the uncertainty level in northern Guangdong is significantly higher compared to other regions, especially in some specific areas where the uncertainty even exceeds 1.5 mm/year, reflecting a relatively high error in the deformation results for these regions. In addition to northern Guangdong, the areas around Leizhou Peninsula, Hanjiang Delta, Yunfu City, and Zhaoqing City also exhibit relatively high uncertainty, further indicating potential errors in the deformation results for these regions. Nonetheless, overall, the uncertainty of the deformation rate in most areas of Guangdong Province remains at a low level, thus ensuring that the accuracy of the deformation measurements meets the needs of our research.

We monitored six mining sites in northern Guangdong (Figure 6a–f) with great deformation velocities. Mining area A is located in the Dabaoshan mining site, with subsidence rates ranging from −40 to −5 mm/year. This region has two large subsidence funnels, with subsidence rates exceeding 40 mm/year, which are attributed to long-term mining activities. Mining sites B to F are also distributed in northern Guangdong, including cities like Shaoguan and Qingyuan.

We analyzed the deformation of three regions (Figure 4a–c) in Guangdong Province. Region a is the Leizhou Peninsula. The northeastern part of Leizhou City and the eastern part of Xuwen County have great subsidence, with regional subsidence rates exceeding 40 mm/year. The subsidence in Leizhou City is founded along the river channels in the eastern part. Optical imagery shows that these regions are agricultural fields, with some coastal mudflats. The maximum subsidence rate in this region exceeds 40 mm/year. A large area of deformation is found in Xuwen County, along the eastern cultivation zones and river channels. Region b is the Pearl River Delta. Surface subsidence is concentrated at the delta front and coastal areas, with subsidence rates ranging from −40 to −5 mm/year, and some areas exceeding −40 mm/year (Figure 4). The delta front, including the southwestern and northern parts of Doumen District in Zhuhai and the western areas of Macau, is heavily affected by subsidence. The coastal areas suffering from significant subsidence are the Maolong Waterway of Cuiheng New Area in Zhongshan City and the northern part of the Gangwan Avenue in Zhongshan City. By selecting the time-series points in the southwest of Doumen District to plot the time series (Figure 7a), it can be observed that the cumulative subsidence in this area reached 28 cm from 2017 to 2022, with an average subsidence rate of −50.25 mm/year. Since 2017, the subsidence has continued without any sign of stopping, indicating that this region will continue to subside in the future. Region c is the Hanjiang Delta, where the largest subsidence area is in Shantou City, especially the southwestern part that has deformation rates ranging from −40 to −10 mm/year (Figure 4). The coastal regions of Shantou City like Jinshi Hou and Maqian have the greatest subsidence rates. Interestingly, in Puning City, some small areas experienced uplift, with the maximum rate reaching 30 mm/year. The time-series deformation at a point in this area is shown in Figure 7b, with a uplift rate of 23.65 mm/year. From early 2017 to the end of 2022, the maximum cumulative uplift exceeded 120 mm, indicating a stable uplift trend.

In the coastal areas, significant deformation occurred in the reclaimed land zones, such as the southern part of Zhanjiang Port in Zhanjiang City, the western side of the Nafu River estuary in Jiangmen City, the southern region to the west of Huangmao Sea, the southern coastal areas of Jinwan District in Zhuhai City, the southern part of Macau, Zhongshan Park and Bancho Reef in Zhuhai, Longxue Island in Nansha District, the eastern coastal areas of Zhuhai Port, the coastal areas near Qianhai Bay and the eastern coastal regions of Lingdingyang in Shenzhen, the northern part of the Dutou River estuary in Huizhou City, and the eastern coastal areas of Maqian and Jinshihou in Shantou City (Figure 8). Some of these areas exhibit a maximum subsidence rate of −40 mm/year or more, including the southern coastal areas of Jinwan District in Zhuhai City, Zhongshan Park and Bancho Reef in Zhuhai, the eastern coastal areas of Zhuhai Port, and the eastern coastal regions of Maqian and Jinshihou in Shantou City. The remaining regions have a small area of subsidence.

To validate the deformation results, this study conducted field investigations in the Pearl River Delta region. Figure 9 shows the photos of subsidence in some regions: a water pump house in Pingsha Town, Jinwan District, Zhuhai City; an elliptical-shaped building in the Fourteenth Village of Tanzhou Town, Zhongshan City; and a stilted house in Ma’an Village, Nanlang Town, Zhongshan City. These photos demonstrate that these buildings have all experienced land subsidence, validating the findings of this study.

## 4. Identification and Statistical Analysis of Surface Deformation

This paper employs a wide-area deformation automatic identification method based on time-series InSAR deformation results to identify surface deformation regions in Guangdong Province and calculate their area. The method consists of two steps: (1) Setting deformation rate thresholds. Points with deformation rates exceeding the threshold are defined as deformation points. The deformation points are analyzed using the spatial clustering algorithm and set the expansion radius. Deformation points within the expansion radius are considered to belong to the same deformation area. This expansion radius is determined based on the region, resolution, and geological conditions. (2) Selecting deformation regions. Some deformation regions identified in step (1) may be caused by noise or errors. A minimum deformation area is set to remove them.

In this study, the deformation rate threshold was set as 15 mm/year, the expansion radius as 120 m, and the minimum area threshold as 0.1 km^2^. Finally, 2394 deformation areas were identified in Guangdong Province, with a total area of 3230.6 km^2^. The number and area of deformation areas in each city are shown in Table 2, and the percentage of deformation area in each city is illustrated in Figure 10.

As Figure 10 and Table 2 show, the deformation areas in Guangdong Province are concentrated in coastal cities, such as Zhanjiang City, Jiangmen City, Zhongshan City, Shantou City, and Zhuhai City, each with the deformation area exceeding 100 km^2^. Zhanjiang City has the most deformation regions, totaling 846, with an area of 1494.8 km^2^, accounting for 46.27% of the total deformation area of the province. We superposed the results on an optical base map (Figure 11a) and found that subsidence is the main geological hazard in this city, affecting a considerable area, especially agricultural lands. Zhanjiang city has high temperatures and evaporation rates. These factors coupled with the limited water storage infrastructure lead to the excessive exploitation of groundwater, resulting in declining groundwater levels and inducing extensive land subsidence. In the Pearl River Delta, the southwest part of Zhuhai City exhibits the most severe and extensive deformation. The areas with deformation rates exceeding 15 mm/year are as large as 405.2 km^2^. Most deformation areas are in the central part of Zhuhai City, exhibiting significant deformation rates (Figure 11b). On the east and west sides of the main deformation area are river channels. The loose geological structure, loading from construction projects, and groundwater extraction together have contributed to large area land subsidence. The deformation regions in Jiangmen City, with an area of 332.2 km^2^ and deformation rates of 15–30 mm/year, are mostly paddy fields and reclamation areas (Figure 11c). The boundary area between Jiangmen City and Zhongshan City exhibits 197 and 128 deformation areas, respectively, which may be related to the soft soil layers. Using the automatic identification method, we identified the areas with the deformation rates exceeding 15 mm/year, 30 mm/year, and 50 mm/year in Guangdong Province separately (Table 3). Most of the deformation regions have deformation rates in the range of 15–30 mm/year. The deformation regions with rates exceeding 50 mm/year cover an area of 65.6 km^2^. Only one deformation region has a rate exceeding 100 mm/year. It is located in Shanwei City and covers an area of 0.2 km^2^. Considering optical imagery, we infer that the deformation in this area is primarily attributed to construction activities.

## 5. Discussion

### 5.1. Temporal and Spatial Characteristics of Surface Deformation in Guangdong Province

The average deformation rate map (Figure 8) and the results of identification and statistical analysis of deformation areas (Figure 9) show that deformation regions in Guangdong Province are concentrated in the Leizhou Peninsula, the Pearl River Delta, the Hanjiang Delta, and coastal regions, and some deformation points are scattered in northern Guangdong, western Guangdong, and eastern Guangdong. The deformation can be classified into four types. Type 1 represents land subsidence in coastal urban areas. Such subsidence mainly occurs in the Leizhou Peninsula, the Pearl River Delta, and the Hanjiang Delta, as well as coastal regions. This subsidence results from the combined contribution of groundwater extraction, building loads, agricultural activities, and soft soil layers. Most subsidence regions have a subsidence rate smaller than 50 mm/year, covering an area of 3789.079 km^2^, accounting for 98.3% of the total subsidence area. A few subsidence regions have rates exceeding 100 mm/year. Type 2 represents subsidence in cities such as Zhuhai, Zhanjiang, and Zhongshan. The subsidence area of each city is larger than 100 km^2^. Zhanjiang City has 781 deformation regions, the most of all cities of this province, accounting for 33.5% of the total number of identified subsidence regions. The total area of these subsidence regions is also the largest among these cities, reaching 1439.3 km^2^, which accounts for 45.3% of the entire subsidence area. This is because this city primarily relies on groundwater for its urban water supply. Type 3 represents uplifts in Puning, with the highest uplift rate of 30 mm/year. The uplift is likely related to the groundwater level rise following the ban of groundwater extraction. Type 4 represents surface deformations due to mining activities in northern cities like Shaoguan, Heyuan, Meizhou, and Qingyuan. The deformation is characterized by great subsidence rates, exceeding 40 mm/year, resulting in subsidence funnels. Additionally, some deformations in western Guangdong and northern Guangdong were caused by geological hazards such as landslides, collapses, and mudslides. Those deformations have a small area and wide distribution, which is challenging to identify.

### 5.2. The Correlation between Geological Hazards and Human Activities

We analyzed the relation between the surface deformations in Guangdong province and the geological structure. We found that the distribution of geological hazards in Guangdong is closely related to the soft soil distribution in the delta regions, hidden karst distribution in the central Guangdong region, ore-forming conditions in western Guangdong, and fault structures. The delta regions have dense river networks. The sediment carried by water deposits along river channels. At the land–sea interface, sediment deposition causes the coastline to expand outward. The accumulation of sediment forms soft soil. Soft soil layers have high porosity, compressibility, and moisture content, but poor permeability, shear strength, and load-bearing capacity. As they are located in low-lying areas along rivers, coasts, and deltas, they are susceptible to ground subsidence under the influence of groundwater level changes, heavy rainfall, coastal erosion, and human engineering activities [30]. The thickness of soft soil layers is strongly correlated with ground subsidence. Regions with thicker soft-soil layers experience more frequent and severe subsidence. As is shown in Figure 10, the deformation regions are concentrated in the delta regions, indicating that the widespread distribution and significant thickness of soft soil layers are the main causes of ground subsidence. Central regions in Guangdong, like Qingyuan and Huadu of Guangzhou, have hidden karst formations. These formations are covered by loose, unconsolidated quaternary sediments of varying thicknesses. The uppermost layers of underground karsts are easily affected by groundwater level changes, leading to the formation of soil cavities, which in turn trigger ground subsidence, collapse, cracks, and other geological hazards [31,32]. In the northern region, there are numerous large mining sites. Mining activities have caused significant surface deformation in this region.

As the coastal region’s economy grows and population rises, urban land resources become increasingly scarce. To accommodate urban expansion, reclamation and land filling that involve transforming water bodies and tidal flats into usable land are adopted [33,34,35,36]. However, the filling material is silt, which has a natural consolidation process. In a short period, the geological structure of the reclaimed region becomes unstable. In addition, this region distributes thick, soft layers, which have low load-bearing capacity, caused by the increased loads from buildings and roads, leading to ground subsidence. Furthermore, self-consolidation of the filling material can also contribute to subsidence. Additionally, excessive extraction of groundwater for agriculture and aquaculture has led to a rapid decline in the water table, reducing pore water pressure in the soft soil layers and increasing effective stress, thereby causing soil compaction. Since soft soil layers are highly compressible, compaction results in ground subsidence.

### 5.3. Sentinel-1A/B Disaster Monitoring Capability and Limitations in Guangdong Province

Sentinel-1A/B images can reveal the evolution characteristics of wide-area land subsidence in Guangdong Province, especially the coastal areas and delta regions [28]. These images can also be used for monitoring large- and medium-area land subsidence associated with human activities such as mining, subway construction, landfilling, and coastal land reclamation [18,20,24]. However, few studies have focused on small-scale deformations, like slope cutting, foundation pits, and individual buildings. One reason is that Sentinel images have a medium resolution (10–20 m), which is not high enough for monitoring small deformations. In such a case, higher-resolution commercial satellite data are required. In addition, Sentinel images have a single observation geometry (ascending orbits in Guangdong Province) that cannot reveal multi-dimensional deformation features [30,32].

Sentinel C-band images also result in difficulties monitoring small-area geological hazards, such as minor landslides, slope failures, and mudslides, that are widespread in the hilly regions of northern and western Guangdong. These geological hazards are characterized by their high density, wide distribution, and small area (90% smaller than 1000 m^2^). Additionally, these hazards often occur during the rainy season from April to August, including the typhoon season. The Sentinel SAR images are affected by severe atmospheric turbulence and phase decorrelation. However, the lush surface vegetation exacerbates phase decorrelation. Fortunately, the short revisit cycle of Sentinel-1A/B (12/24 days) and long-wavelength L-band NISAR satellites can reduce the effects of phase decorrelation. However, the temporal distribution of the data can be limited because of the unavailability of Sentinel-1B due to technical problems.

In this study, the deformations in western and northern Guangdong are primarily related to agricultural areas. The network of the spatial–temporal baseline and multi-look processing significantly determine the result of deformation. Deformation in hilly mountainous areas may contain atmospheric delay errors and phase decorrelation, posing challenges for the identification of geological hazards. In provinces like Sichuan, Guizhou, and Yunnan, landslides occur on a larger scale and surface signals are more pronounced. Combining artificial intelligence and visual interpretation can yield better identification accuracy [36,37].

In summary, C-band Sentinel images are effective in monitoring wide-area ground subsidence and collapses in coastal delta regions, providing valuable insights for analyzing geological hazard causes and mitigation measurements. However, for small-area landslides, avalanches, and engineering-specific geological hazards, the monitoring effectiveness of Sentinel images is limited. This is due to the single observation geometry, medium spatial resolution, and susceptibility to dense surface vegetation cover. In June 2023, China launched the L-band Land Survey-1 (Lutan-1) satellite. With its all-day monitoring capability and 3 m spatial resolution at 28-day intervals, Lutan-1 offers a partial solution for wide-area InSAR geological hazard surveys in Guangdong Province [36,37]. Furthermore, the open-access L-band NISAR data can provide abundant observations for further geological hazard surveys. By combining medium-resolution C-band Sentinel images with high-resolution L-band Lutan-1 and NISAR data, comprehensive geological hazard surveys and monitoring can be achieved in Guangdong Province, utilizing multiple geometries and observation platforms.

## 6. Conclusions

This paper utilized the 2015–2022 Sentinel-1A/B data to obtain high-resolution (40 m) surface deformation data in Guangdong Province by MT-InSAR (Multi-Temporal Interferometric Synthetic Aperture Radar) technology. We analyzed the deformation results of the mining areas in northern Guangdong, the delta regions, and the land reclamation areas along the coast. Additionally, we conducted surface deformation region identification and area counting for each city in Guangdong Province, and obtained the following findings: (1) Surface deformation is concentrated in the coastal urban areas of the deltas; the mining areas in northern Guangdong; regions affected by landslides, avalanches, and mudslides; as well as the coastal land reclamation areas. (2) Using an automated identification method, we identified a total of 2394 deformation regions in Guangdong Province, covering 3230.6 km^2^, and regions with deformation rates exceeding 15 mm/year covering 3175 km^2^. (3) By exploring the relationship between geological hazards and human activities, we found that geological hazards are contributed to by land reclamation, seawater filling, groundwater extraction, and various construction activities, influenced by geological factors such as soft soils in the delta regions, concealed karst distribution in central Guangdong, mining conditions in western Guangdong, and fault structures. These findings have significant implications for geological hazard surveys and mitigation efforts in Guangdong Province.

## Figures and Tables

**Figure 1 sensors-24-05449-f001:**
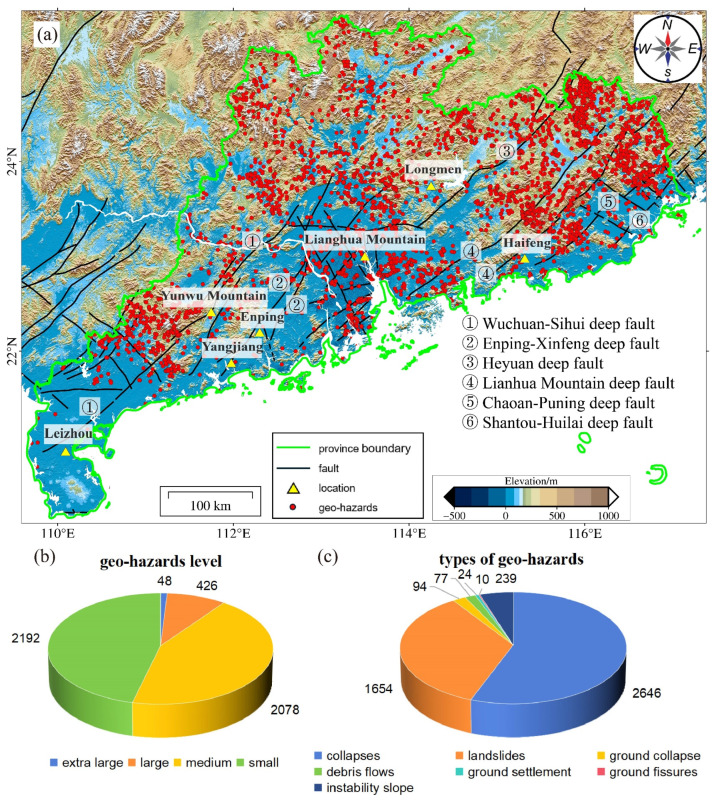
The spatial distribution, level, and types of geological hazards in Guangdong Province. (**a**) Geological background and disaster distribution of the study area on a color-shaded elevation map; (**b**) the pie chart of geological hazard level; (**c**) the pie chart of geological hazard types.

**Figure 2 sensors-24-05449-f002:**
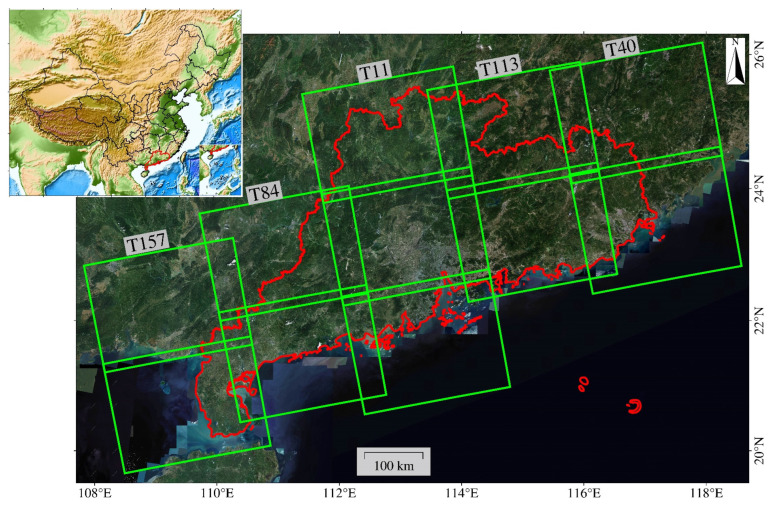
Coverage of the Sentinel-1 images over Guangdong Province from 2015 to 2022 including 11 frames in 5 tracks. The green line presents the footprint of Sentinel 1A/B data and the red line presents the boundary of Guangdong Province.

**Figure 3 sensors-24-05449-f003:**
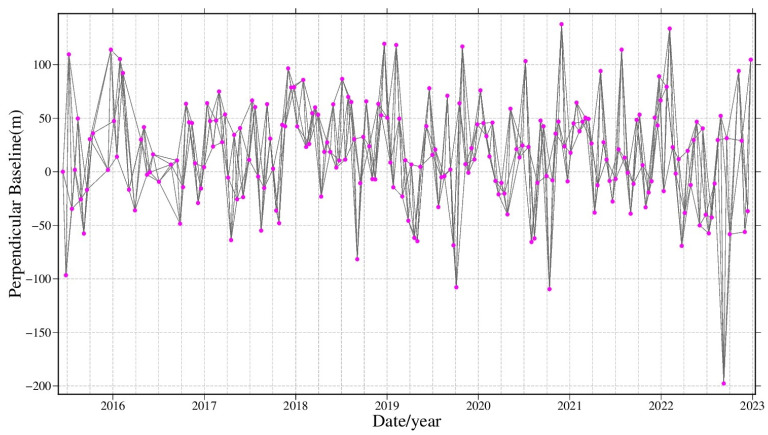
The interferogram network figures are based on spatial–temporal baselines.

**Figure 4 sensors-24-05449-f004:**
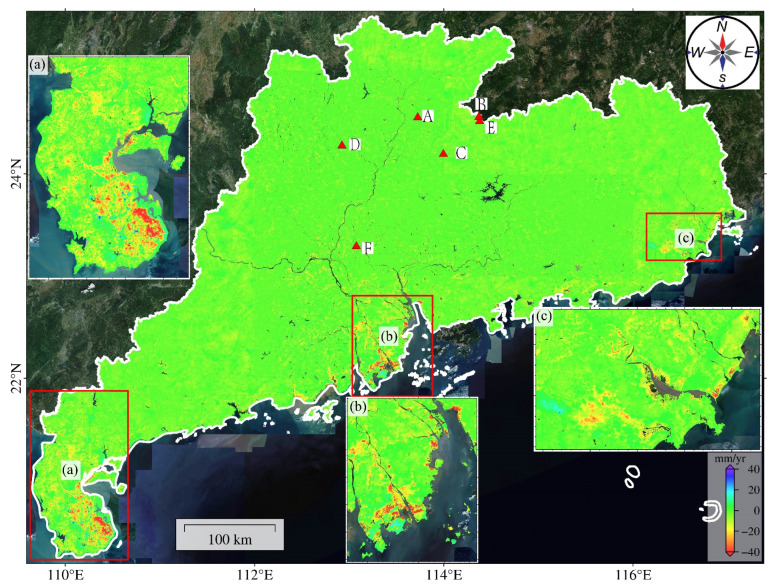
Surface deformation velocity map of Guangdong Province for the period 2015–2022. A–F are six mining sites with large deformation; (**a**–**c**) are the zoom-ins of the Leizhou Peninsula, Pearl River Delta, and Hanjiang Delta.

**Figure 5 sensors-24-05449-f005:**
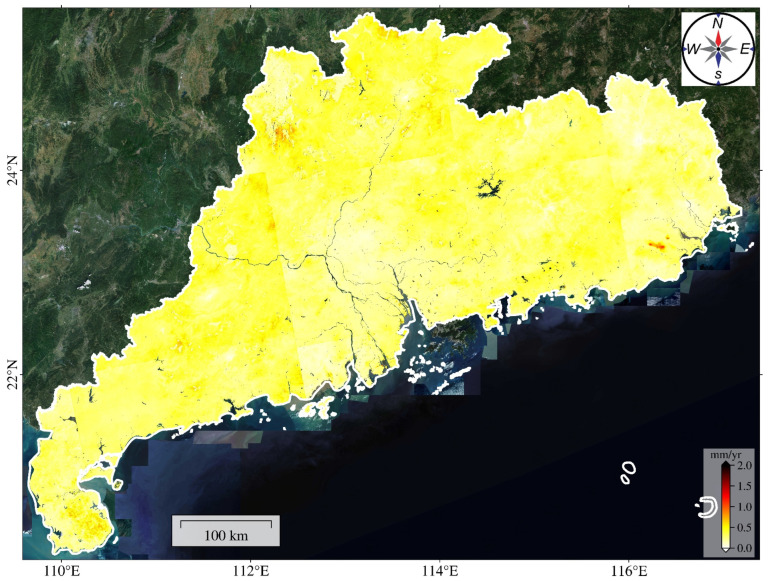
The uncertainty of the surface deformation velocity map of Guangdong Province for the period 2015–2022.

**Figure 6 sensors-24-05449-f006:**
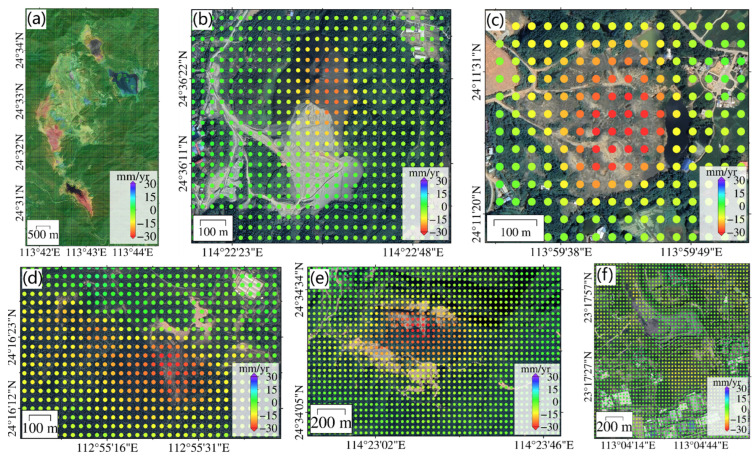
Surface deformation velocity of the six selected mining areas in northern Guangdong Province. The figures (**a**–**f**) correspond to the six selected mining areas A–F in Figure 4.

**Figure 7 sensors-24-05449-f007:**
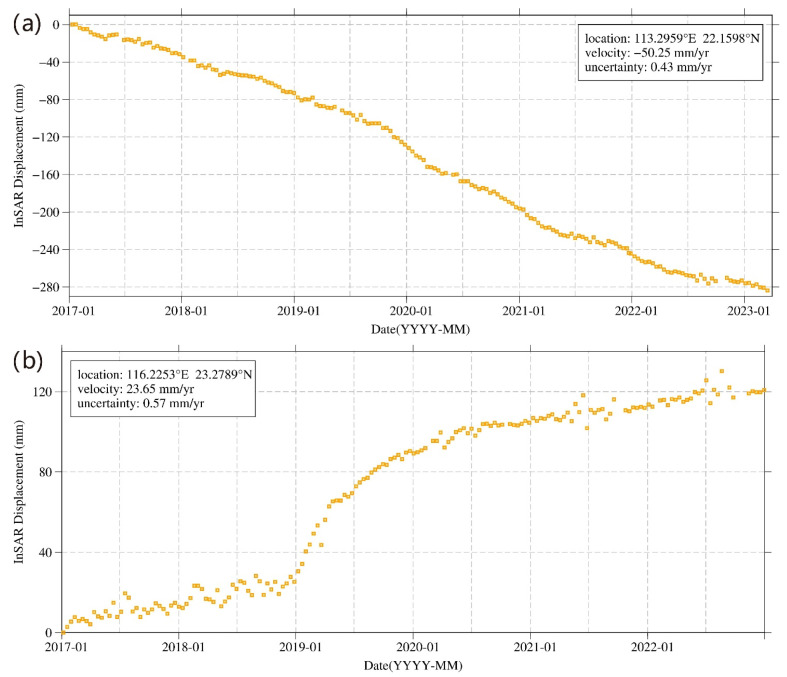
Surface deformation sequence results of (**a**) the Zhuhai reclamation area and (**b**) the uplift region of Puning, Jieyang.

**Figure 8 sensors-24-05449-f008:**
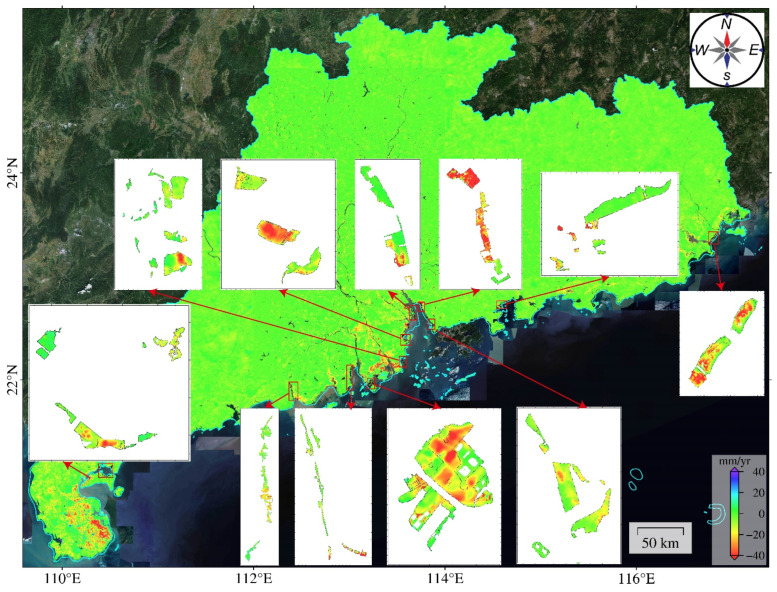
Surface deformation rate map of coastal reclamation areas in Guangdong Province.

**Figure 9 sensors-24-05449-f009:**
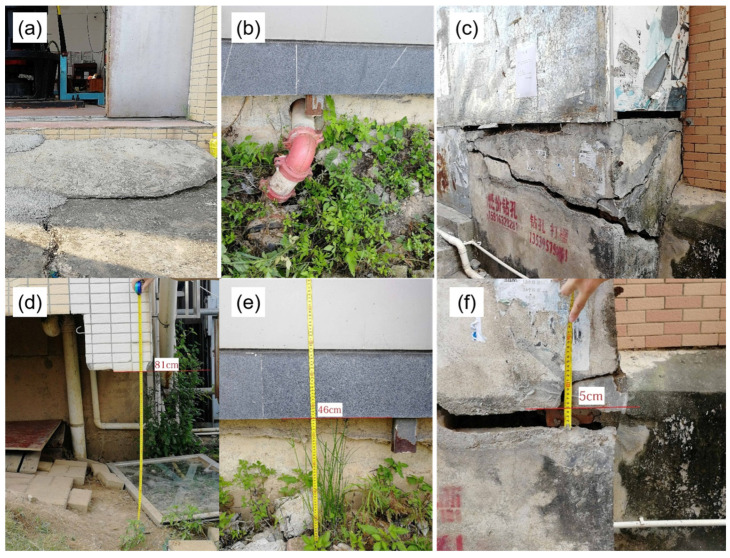
Field survey photos of ground subsidence in Zhuhai. (**a**,**b**) A water pump house in Pingsha Town, Jinwan District, Zhuhai City; (**c**,**d**) an elliptical-shaped building in the Fourteenth Village of Tanzhou Town, Zhongshan City; (**e**,**f**) a stilted house in Ma’an Village, Nanlang Town, Zhongshan City.

**Figure 10 sensors-24-05449-f010:**
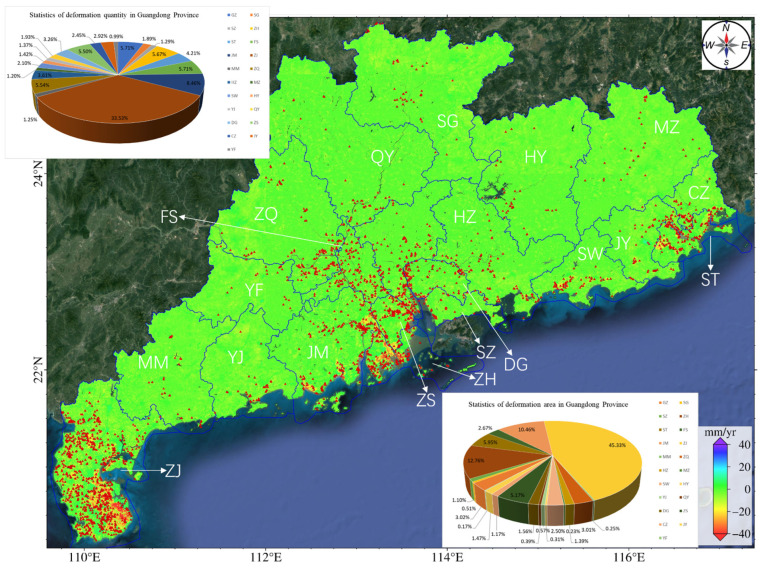
Deformation zones in Guangdong Province were identified by the automatic deformation detection method. The red points denote the location of deformation areas, and the blue lines are the administrative boundary.

**Figure 11 sensors-24-05449-f011:**
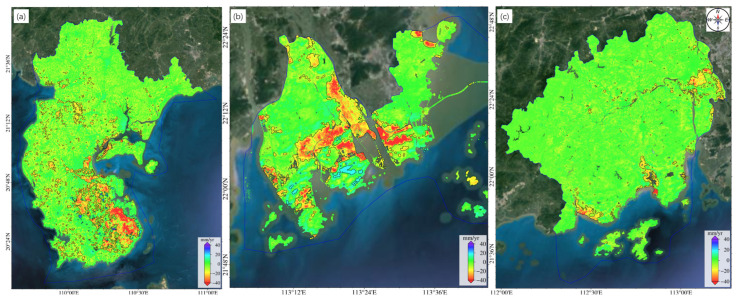
Identified deformation zones in (**a**) Zhanjiang City, (**b**) Zhuhai City, and (**c**) Jiangmen City.

**Table 1 sensors-24-05449-t001:** Sentinel-1A/B satellite parameters and data list.

Satellite Sensor	Parameters	Value
Sentinel-1A/B	Wavelength	5.6 cm (C-band)
	Flight direction	Ascending
	Type of product	SLC
	Revisit cycle	12/24 days
	Image mode	IW
	Polarization mode	VV
	Incidence angle	29°–46°
	Azimuth angle	348.5°–350.3°
	Acquisition time	2015/06–2022/12

**Table 2 sensors-24-05449-t002:** Identified surface deformation region information in each city of Guangdong Province.

City	Number	Number Percentage (%)	Area (km^2^)	Area Percentage (%)
Guangzhou (GZ)	133	5.71%	96.0	3.02%
Shaoguan (SG)	44	1.89%	16.1	0.51%
Shenzhen (SZ)	30	1.29%	35.0	1.10%
Zhuhai (ZH)	132	5.67%	405.2	12.76%
Shantou (ST)	98	4.21%	188.8	5.95%
Foshan (FS)	133	5.71%	84.9	2.67%
Jiangmen (JM)	197	8.46%	332.2	10.46%
Zhanjiang (ZJ)	781	33.53%	1439.3	45.33%
Maoming (MM)	29	1.25%	7.8	0.25%
Zhaoqing (ZQ)	129	5.54%	95.6	3.01%
Huizhou (HZ)	84	3.61%	44.1	1.39%
Meizhou (MZ)	28	1.20%	7.2	0.23%
Shanwei (SW)	49	2.10%	79.3	2.50%
Heyuan (HY)	33	1.42%	9.9	0.31%
Yangjiang (YJ)	32	1.37%	18.0	0.57%
Qingyuan (QY)	45	1.93%	12.5	0.39%
Dongguan (DG)	76	3.26%	49.7	1.56%
Zhongshan (ZS)	128	5.50%	164.07	5.17%
Chaozhou (CZ)	57	2.45%	37.2	1.17%
Jieyang (JY)	68	2.92%	46.6	1.47%
Yunfu (YF)	23	0.99%	5.4	0.17%

**Table 3 sensors-24-05449-t003:** The areas of the deformation regions with different deformation rates in Guangdong Province.

Deformation Velocity	Area (km^2^)
$\left|vel\right|\geq 15\;mm/year$	3175.0
$\left|vel\right|\geq 30\;mm/year$	614.1
$\left|vel\right|\geq 50\;mm/year$	65.6
$\left|vel\right|\geq 100\;mm/year$	0.2

## Data Availability

The Sentinel-1A/B data used in this study are copyrighted by the European Space Agency (https://search.asf.alaska.edu/, accessed on 1 April 2023).

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
