# Peer review of "Non-Tectonic Geohazards of Guangdong Province, China, Monitored Using Sentinel-1A/B from 2015 to 2022"

_sensors, 2024, doi:10.3390/s24165449_

Round 1

Reviewer 1 Report

Comments and Suggestions for Authors

Thank you very much for inviting me to review this work titled "Geo-hazards of Guangdong Province, China, monitored using Sentinel-1A/B from 2015-2022." First of all, I would like to congratulate the authors for their work, as it shows significant effort in processing so many radar images over such a large area. I have reviewed the paper and attached a PDF with the main comments I have found.

I have two major concerns about this work:

  • The use of the term "Velocity" without specifying which reference system is being used. I understand that the authors are referring to velocity in the satellite's LOS vector, but in the text, they directly relate subsidence to the LOS vector. This can be misleading, as a reader will associate subsidence with the vertical component of the displacement, and when working with LOS and full-frame images, we will have significant variations in the incidence angle. On the other hand, if they have performed some kind of projection, they should specify how it was done. In my case, for a mining area, I performed a decomposition into the vertical component based on a critical angle and below this in the direction of maximum slope (similar to what is done with landslides). There are many ways to do it from many authors, you can check Escayo et al 2022 article about Riotinto mine or any other with this kind of decomposition.
  • The figures are too small, I recommend to improve the readability of them in order to be bigger.

I strongly recommend that the authors review these points and resubmit the article, I think those are not major issues to fix.

Author Response

Response to Reviewer 1 Comments

Notes:

(1) black bold fonts indicate the comments from reviewer;

(2) blue fonts indicate the authors’ response;

(3) black italic fonts indicate the text in the revised manuscript without modification traces.

Comments from Reviewer 1:

(1). In line 13,Guangdong Province in China has many cities with dense population and developed economy.’, Maybe this sentence is too vague, could you give an estimation?

>> Thank you for your comments. I have provided a quantitative description of the population, cities, and economy of Guangdong Province.

In lines 13-14:

Guangdong Province, home to 21 cities and a permanent population of 127.06 million, boasts the largest provincial economy in China, contributing 11.76% to the national GDP in 2023.

(2). In line 25, It is better to use manuscript or current work than "paper"

>> Thank you for your good suggestion. Corrected as suggested.

In line 25:

To address these issues, current work utilizes Sentinel-1A/B satellite data covering Guangdong Province from 2015 to 2022 to obtain the wide area surface deformation in the whole province by the multi-temporal (MT) InSAR technology.

(3). In line 31, "(1) Using the automatic identification algorithm we identified 2394 defamation regions;", the "defamation" is not correct in this context.

>> Thank you for your comments. Corrected as suggested.

In line 31:

(1) Using the automatic identification algorithm we identified 2394 deformation regions;

(4). In line 33, "Great subsidence velocity is observed in the hilly regions of northern Guangdong, particularly in mining areas; ",Use ratio instead of velocity and avoid "Great", instead use a value or a threshold like "over 1 cm/y"

>> Thank you for your comments. Corrected as suggested.

In line 32:

Over 4cm/yr subsidence rate is observed in the hilly regions of northern Guangdong, particularly in mining areas;

(5). In lines 54-55, a reference to the related studies should be added.

>> Thank you for your good suggestion. I have added the related reference in this point.

Revises text in lines 54-55:

These hazards have caused economic loss as huge as 8.18 billion RMB and affected the life of 249,500 people [2].

Reference:

  1. Tong, J. Analysis of Development Characteristics and Influencing Factors of Geological Hazards During the "Thirteenth Five-Year Plan" Period in Guangdong Province. Mod. Min. 2021, 37, 219-221.

(6). In lines 70, 98, 107, 113, 116, 129, 138, Redundant hyphens exist.

>> Thank you for your comments. Corrected as suggested.

In lines 71, 76, 98, 107, 116, 129, 141

(7). In lines 154, a reference to the related studies should be added.

>> Thank you for your comments. I have added the related reference in this point.

Revises text in line 156:

According to official records [2], the province has 18 high-risk geological hazard areas and 17 moderate-risk areas, covering an area of 104,600 square kilometers, which accounts for 58.1% of the total land area of the province (Figure 1(a)).

  1. Tong, J. Analysis of Development Characteristics and Influencing Factors of Geological Hazards During the "Thirteenth Five-Year Plan" Period in Guangdong Province. Mod. Min. 2021, 37, 219-221.

(8). In Figure1, the color schema used in Figure1 can be difficult to read for colorblind persons.

>> Thank you for your comments. Figure 1 contains abundant background information, so using different colours can help distinguish between different information. In the study area, most regions have the elevation below 500. To improve the quality, we have reexported the figure with higher resolution.

(9) In Table1, the line 187, the column of “Image Number” in Table1 is unnecessary, it is suggested to remove, the column of "Acquisition Time" can be included as another row in parameter's column instead to a new column.

>> Thank you for your comments. Corrected as suggested.

In line 185:

Table 1. Sentinel-1A/B satellite parameters and data list.

Satellite Sensor

Parameters

Values

Sentinel-1A/B

Wavelength

5.6cm(C-band)

Flight direction

Ascending

Type of product

SLC

Revisit cycle

12/24 days

Image mode

IW

Polarization mode

VV

Incidence Angle

29°-46°

Azimuth angle

348.5°-350.3°

Acquisition Time

2015/06-2022/12

(10) In Table1, the line 187, it should be commented in the text that you use ascending orbit because the lack of images in descending orbit.

>> Thank you for your comments. Corrected as suggested.

In lines 179-180:

To obtain the wide-area surface deformation of Guangdong Province, we collected the SAR images covering the study area acquired by C-band Sentinel-1 satellite from June 2015 to December 2022. Due to the lack of the descending orbit Sentinel-1 data in Guangdong Province, we only acquired the ascending orbit data.

(11) In line 192, the software you use for this processing didn't mentioned. I think that this information is mandatory in any InSAR paper in order to ensure the reproducibility of the results.

>> Thank you for your comments. We used GAMMA soft to process the data. I have added the description of this point.

In line 191:

3.2. Multi-Temporal InSAR Technology

The study area is calculated by the GAMMA software. We used a time-series SBAS InSAR technique based on multi-master images to obtain the surface deformation in Guangdong Province from June 2015 to December 2022.

(12) In line 198, Usually a 0.001 pixel accuracy is required for TOPS, are you sure that you used 0.003?

>> Thank you for your comments. I am sorry for the mistake. The TOPS imaging mode is the default operation mode of Sentinel-1 satellite, requiring an overall azimuth co-registration accuracy better than 0.001 pixels. We have corrected this mistake.

In 196:

The data processing procedure is as follows: (1)Data pre-processing. First, each frame's images were registered with a registration accuracy better than 0.001 pixels.

(13) In lines 204-205, you also can add that this kind of multilook provides a ground resolution of about 40x40 meters (which is also in table1).

>> Thank you for your suggestion. We have added this description as suggested.

In line 204:

The selected interferometric pairs were processed with a differential interferometry ratio of range to azimuth of 10:2 (ground resolution of about 40x40 meters) to reduce noise and data volume.

(14). The use of the term "Velocity" without specifying which reference system is being used. I understand that the authors are referring to velocity in the satellite's LOS vector, but in the text, they directly relate subsidence to the LOS vector. This can be misleading, as a reader will associate subsidence with the vertical component of the displacement, and when working with LOS and full-frame images, we will have significant variations in the incidence angle. On the other hand, if they have performed some kind of projection, they should specify how it was done. In my case, for a mining area, I performed a decomposition into the vertical component based on a critical angle and below this in the direction of maximum slope (similar to what is done with landslides). There are many ways to do it from many authors, you can check Escayo et al 2022 article about Riotinto mine or any other with this kind of decomposition.

>> Thank you for your good suggestion and comment. I am sorry for my ambiguity expression. The results displayed were projected to the vertical direction. There are 5 path and 11 frame datasets used in this study. In order to obtain the consistent result, we have performed a decomposition to obtain vertical component based on a variable critical angle. We have added related description to the updated manuscript.

In lines 231-233:

Using the method in section 3.2, we obtain the deformation velocity map in every path and frame. In order to make the surface deformation have the same project system, we convert the LOS deformation into vertical direction [30].

Added reference

[30]Escayo, J.; Marzan, I.; Martí, D.; Tornos, F.; Farci, A.; Schimmel, M.; Carbonell, R.; Fernández, J. Radar Interferometry as a Monitoring Tool for an Active Mining Area Using Sentinel-1 C-Band Data, Case Study of Riotinto Mine. Remote Sens. 2022, 14, 3061. https://doi.org/10.3390/rs14133061

(15). In Figure6, the figures are too small, I recommend to improve the readability of them in order to be bigger.

>> Thank you for your good suggestion. We have improved the quality and readability of figures in the updated manuscript.

In figure 6, in lines 270-272:

Figure 6. Surface deformation velocity of the six selected mining areas in northern Guangdong Province.

(16). The Labels of Figure 6,10,11 are too small, cannot read it.

>> Thank you for your good suggestion. I have enlarged the labels of Figure 6,10,11 in the updated manuscript. We also have improved the quality of those figures.

(17). In line 367, Use a quantitative estimation. "A lot of " is not precise.

>> Thank you for your comments. I have replaced "a lot of" with specific quantity.

In lines 369-370:

The boundary area between Jiangmen City and Zhongshan City exhibits significant 197 and 128 deformation areas, respectively, which may be related to the soft soil layers.

(18). In line 379, "5. Discussion" put in a new page.

>> Thank you for your comments. Corrected as suggested.

In line 381

(19). In line 442, it could be good to mention the current unavailability of Sentinel-1B due to technical problems. Since you use it in this study it can affect to the temporal distribution of the data.

>> Thank you for your good comment and reminder. We have added it in line 461-466, section 5.3.

In lines 461-466:

Fortunately, the short revisit cycle of Sentinel-1A/B (12/24 days) satellites can reduce the effects of phase decorrelation, although the temporal distribution of the data can be limited because of unavailability of Sentinel-1B due to technical problems.

(20). In lines 446-447, Sentinel-1 proved to be enough to study this site in many cases, depending on the size. The references do not seem to be very relevant to this statement.

>> Thank you for your comments. We intended to cite literature to demonstrate that Sentinel-1 data, is primarily used for monitoring deformation over larger areas, due to its spatial resolution limitations. The studies focusing on small-scale deformation are relatively rare. Corresponding revisions have been made.

In lines 445-449:

These images can also be used for monitoring large and medium area land subsidence associated with human activities such as mining, subway construction, landfilling, and coastal land reclamation [19,21,25]. However, few studies focusing on small-scale deformation, like slope cutting, foundation pits, individual buildings, remains challenging.

(21). In lines 459-460, it is a bit confusing. In the previous sentences you are talking about the decorrelation caused by the vegetation and the atmosphere. I think that there are other factors that can provide a better improvement over the quality of the results, like using a longer wavelength sensor (like NISAR with an L band sensor), or use a more advanced atmospheric filtering like ERA-5 models (see MintPy's implementation). Of course that Sentinel-1 revisit time is very good, but for a non-expert reader can be a little bit confusing.

>> Thank you for your good comments. This sentence has been rephrased as follows:

In lines 461-464:

Fortunately, the short revisit cycle of Sentinel-1A/B (12/24 days) and long wavelength L-band NISAR satellites can reduce the effects of phase decorrelation, the temporal distribution of the data can be limited because of unavailability of Sentinel-1B due to technical problems.

(22). In lines 462-463, "This is somewhat associated with short baseline networks and multi-look operation." The statement of the sentence is hard to understand.

>> Thank you for your comments. I have revised as below.

In lines 466-467:

The network of spatial-temporal baseline and the processing of multi-look are significant determinant of the result of deformation.

(23). In lines 473- 479, It is very good that talking about Lutan-1 Satellite, but in my experience working with Chinese data is very difficult because there are many restrictions to obtain such data and it is not useful for non-Chinese people. I strongly suggest you to also talk about NISAR which is going to have open-access policy on data (similar to Sentinel-1).

>> Thank you for your suggestion. I have added related expression in our new manuscript.

In line 481-486:

Furthermore, the open access L-band NISAR data also can provide much abundant observation for further geological hazard surveys. By combining medium-resolution C-band Sentinel images with high-resolution L-band Lutan-1 and NISAR data, comprehensive geological hazard surveys and monitoring can be achieved in Guangdong Province, utilizing multiple geometries and observation platforms.

(24). In line 481, "6. Conclusions" put in a new page.

>> Thank you for your comments. Corrected as suggested.

In line 487.

(25). In line 482, "obtain high resolution (20m)", I think this is not correct since you applied a multi-look that provides 40-meter resolution, right?

>>You are right. It is a mistake in this point. The spatial resolution of result obtained in this study is 40 m. we have corrected this mistake in our new manuscript.

In line 488:

This paper utilized the 2015-2022 Sentinel-1A/B data to obtain high resolution (40m) surface deformation data in Guangdong Province by MT-InSAR (Multi-Temporal Interferometric Synthetic Aperture Radar) technology.

Reviewer 2 Report

Comments and Suggestions for Authors

The authors use Sentinel-1A/B data from 2015 to 2022 to conduct a comprehensive survey of the geological hazards throughout Guangdong Province and analyze their causes. I think their results are robust on the conclusions. I only have several minor comments:

1) In lines 68-69, the full names of these abbreviations should be given for the first time, e.g. "PS-InSAR QPS-InSAR, SBAS-InSAR and DS-InSAR".

2) In line 83, a reference to the related studies should be added.

3) In line 123, please replace plateaus with mountains.

4) Is the “sea-son” in line 129 is “season”? Please check it.

5) The statements in lines 127 to 128 are proposed to be changed to read as follows: “The annual average temperature is high, with rainfall concentrated from April to October, coinciding with the highest temperatures."

6) Figure 1 is quoted in the text at line 131, it is suggested that Figure 1 should be moved to the end of section 2.1.

7) Delete plateaus in line 138.

8) Figure 3 Years for which dates need to be added to the horizontal axis.

9) In line 219 you have not stated the method of removing atmospheric errors? Please give the method of removing atmospheric error and literature citation.

10) The colour bar in Figure 6 is not enough to show the size of the deformation in this area, please change the colour bar to make the deformation area more visible.

11) In Section 3.2, some of the parameters of interferogram processing that should be given during time-series InSAR processing are described, such as the coherence threshold during de-entanglement, and the temporal and spatial baseline thresholds at which you screen interferometric image pairs.

12) Section 5.1 analyses and discusses the impact of groundwater on surface subsidence, and whether the relationship can be illustrated by comparing groundwater data changes in a particular city with the InSAR timescale results. For example, Shantou, the city with the most significant degormtion rate.

Round 2

Reviewer 1 Report

Comments and Suggestions for Authors

The authors have done a good job addressing each of the comments that arose in the previous review, and in my opinion, the paper has improved and can be published in the journal in its current form. I would like to congratulate the authors on their work; the task of processing such a large amount of data is immense, and the work shows interesting results for various areas that can be studied in detail in future research.

Author Response

Comment 1: [The authors have done a good job addressing each of the comments that arose in the previous review, and in my opinion, the paper has improved and can be published in the journal in its current form. I would like to congratulate the authors on their work; the task of processing such a large amount of data is immense, and the work shows interesting results for various areas that can be studied in detail in future research.]

Response 1: [Thank you for your positive evaluation and encouragement. We appreciate your thorough review and are delighted that you find our revised manuscript suitable for publication. Your support is invaluable. In the future, we'll continue to conduct more detailed studies in this area.]